# Inflammation-Modulating Hydrogels for Osteoarthritis Cartilage Tissue Engineering

**DOI:** 10.3390/cells9020419

**Published:** 2020-02-12

**Authors:** Rachel H. Koh, Yinji Jin, Jisoo Kim, Nathaniel S. Hwang

**Affiliations:** 1School of Chemical and Biological Engineering, Institute of Chemical Processes, Seoul National University, Seoul 08826, Korea; rhk2115@snu.ac.kr (R.H.K.); enhee616@gmail.com (Y.J.); js8257@snu.ac.kr (J.K.); 2BioMAX/N-BIO Institute, Seoul National University, Seoul 08826, Korea; 3Interdisciplinary Program in Bioengineering, Seoul National University, Seoul 08826, Korea

**Keywords:** immune-modulation, anti-inflammatory, injectable hydrogel, drug delivery, cartilage, tissue engineering

## Abstract

Osteoarthritis (OA) is the most common form of the joint disease associated with age, obesity, and traumatic injury. It is a disabling degenerative disease that affects synovial joints and leads to cartilage deterioration. Despite the prevalence of this disease, the understanding of OA pathophysiology is still incomplete. However, the onset and progression of OA are heavily associated with the inflammation of the joint. Therefore, studies on OA treatment have sought to intra-articularly deliver anti-inflammatory drugs, proteins, genes, or cells to locally control inflammation in OA joints. These therapeutics have been delivered alone or increasingly, in delivery vehicles for sustained release. The use of hydrogels in OA treatment can extend beyond the delivery of anti-inflammatory components to have inherent immunomodulatory function via regulating immune cell polarization and activity. Currently, such immunomodulatory biomaterials are being developed for other applications, which can be translated into OA therapy. Moreover, anabolic and proliferative levels of OA chondrocytes are low, except initially, when chondrocytes temporarily increase anabolism and proliferation in response to structural changes in their extracellular environment. Therefore, treatments need to restore matrix protein synthesis and proliferation to healthy levels to reverse OA-induced damage. In conjugation with injectable and/or adhesive hydrogels that promote cartilage tissue regeneration, immunomodulatory tissue engineering solutions will have robust potential in OA treatment. This review describes the disease, its current and future immunomodulatory therapies as well as cartilage-regenerative injectable and adhesive hydrogels.

## 1. Pathological Condition of Osteoarthritis

### 1.1. Social Burden of Osteoarthritis (OA)

Articular cartilage, which covers the end of each bone, acts as a cushion between the bones and provides a smooth, gliding surface for joint motion. Osteoarthritis (OA) is a joint disease in which the protective cushion breaks down and becomes thinner, initially causing pain, stiffness, and swelling. In severe cases, the cartilage completely wears down and the bones rub against one another, causing severe pain, inflammation, and eventually, immobility.

The risk factors associated with OA can be classified into two groups: primary (non-traumatic) and secondary (traumatic) risk factors [1]. Primary risk factors include age, gender, family history, and obesity. Secondary risk factors involve traumatic injuries, mechanical deformities, and joint laxity [2]. Although OA has a multifactorial etiology and can be considered the product of an interplay between primary and secondary factors, age and obesity are key risk factors that predispose patients to wear and tear of the cartilage and altered joint biomechanics. Particularly, these risk factors play roles in OA development and progression in weight-bearing joints such as the knee. Therefore, knee OA is more common than OA of other joints [3].

OA is one of the most common musculoskeletal diseases that afflicts 240 million people worldwide [4]. In the 2010 World Health Organization Global Burden of Diseases Study, OA was ranked as 11th among 291 conditions that are not fatal, but lead to long years lived with disability (YLD) [5]. A survey of 10,000 patients afflicted with OA reported that 81.5% of the patients experienced limitations in their daily activities and work [6]. OA is increasingly prevalent in the aged and obese populations, the same population that is afflicted with adult diseases such as hypertension and diabetes mellitus [7]. These comorbidities add considerably to the costs and complexity of treatment of patients with OA. The burden of OA is expected to further increase as the world population is increasingly aging and becoming obese. The clinical approaches to OA are expected to change considerably and research on effective OA treatments should be actively conducted.

### 1.2. Pathogenesis of OA

Traditionally, OA has been regarded only as an age-related degenerative disease, but recently, OA has been reclassified as an inflammatory systemic disease with abnormal metabolic overtones of the resident cells called chondrocytes. Articular cartilage consists of chondrocytes surrounded by a dense extracellular matrix (ECM) network of mainly proteoglycans and collagen. In normal joints, chondrocytes maintain joint homeostasis by modulating ECM synthesis and degradation. In osteoarthritic joints, pro-inflammatory cytokines such as interleukin 1-beta (IL-1β) and tumor necrosis factor (TNF-α) activate the NF-κB signaling pathway that causes systemic inflammation of all joint tissues including the cartilage, synovial membrane, subchondral bone, and ligaments (Figure 1) [8,9]. Chondrocytes particularly experience great stress in the inflammatory environment and undergo various phenotypical changes: elevated expression levels of pro-inflammatory cytokines (TNF-α, IL-1β, and IL-6) and ECM degradative enzymes (matrix metalloproteinases (MMPs) and aggrecanases) [10,11], and significantly lower syntheses of ECM molecules. Such phenotypical changes result in net cartilage destruction. The NF-κB molecule also activates nitric oxide (NO), cyclooxygenase 2 (COX-2), nitric oxide synthase (NOS), and prostaglandin E2 (PGE2), which promote catabolism and chondrocyte apoptosis [12]. 

### 1.3. Current Strategies for OA Treatment

#### 1.3.1. Pharmacological Treatments

OA treatments are primarily focused on drugs that reduce pain and inflammation. Corticosteroid drugs (CSDs) and non-steroid anti-inflammatory drugs (NSAIDs) are the most commonly used medications. NSAIDs help reduce pain, decrease fever, prevent blood clots, and decrease inflammation. NSAIDs may have a side effect if taken long-term including gastrointestinal ulcers and bleeds, heart attack, and kidney disease [13]. NSAIDs exhibit anti-inflammatory and antipyretic effects by blocking prostanoid production from arachidonic acid via cyclooxygenase enzyme inhibition [14]. CSDs are powerful anti-inflammatory medicines, but as a type of steroid, they have a wide range of side effects such as the increased risk of hypertension, diabetes, and osteoporosis [15]. 

The most common form of CSD and NSAID administration is oral. Orally administered therapeutic agents take 5–6 h to reach the intra-articular site of action via systemic circulation [16]. The drug concentration in the synovial fluid is an important determinant of the clinical efficacy of the drug [17]. However, the drug concentration is significantly reduced in systemic delivery. Direct intra-articular injection of the drugs can overcome the problem of low drug concentration from oral administration; only a minimal amount of drug is required to exert the desired pharmacological activity. However, rapid clearance of the injected drugs from the synovial fluid is a hurdle to effective treatment. Generally, duration of the soluble drug in synovial fluid is only a few hours after intra-articular injection [18]. Maintaining therapeutic drug concentration over a prolonged time can be achieved by repeated intra-articular administration or more ideally, by immobilization to injectable depot formulations for sustained release.

#### 1.3.2. Viscosupplementation

Viscosupplementation (VS) via intra-articular injection of hyaluronic acid (HA) or lubricin is widely used for symptomatic knee OA. HA is the most commonly used biomaterial for VS. HA is a major component of synovial fluid and cartilage ECM. Intra-articular injection of HA provides symptom relief by enhancing the synthesis of extracellular matrix proteins, regulating inflammatory mediators to shift away from degradation, reducing joint friction, and maintaining cartilage thickness and surface smoothness [19]. Many studies have shown that HA has a greater effect on pain relief than the placebo [20,21]. OA patients with intra-articular injections of HA every six months showed better knee cartilage preservation compared to those without HA injections [22]. 

In the body, HA is subjected to various degradation processes due to hydrolysis and enzymatic degradation by naturally occurring hyaluronidase. The turnover of HA in a joint is surprisingly rapid [23]. Therefore, VS strategies require improved mechanical properties and degradation rates of HA for prolonged treatment effects.

#### 1.3.3. Cell-Based Treatments

Compared to pharmacological and VS treatments that control symptoms, cell-based treatments are more focused on cartilage regeneration. Stem, progenitor, or primary cells are utilized to replace or repair OA-damaged tissues (Figure 2) [24]. The effect of cell-based therapy can be realized through two mechanisms: (1) cell engraftment to the damaged site, and (2) release of regenerative trophic factors.

The basic cell-based cartilage treatment is autologous chondrocyte implantation (ACI). Isolated autologous chondrocytes from non-weight bearing cartilage regions are expanded in vitro and implanted into the affected area [25]. Carticel^®^ (Genzyme) is an FDA-approved ACI treatment [26]. Despite encouraging clinical results, there are still limitations to ACI related to the complexity and cost of two surgical procedures, and the de-differentiation and consequent capacity loss associated with in vitro expansion of isolated chondrocytes [27]. Furthermore, OA is considered a contraindication for ACI in certain countries because ACI is not suitable for widespread cartilage damage, as in the case of osteoarthritis.

Mesenchymal stem cells (MSCs) are considered an excellent alternative cell source to chondrocytes for cartilage regeneration. As adult stem cells capable of chondrogenic differentiation, MSCs have been combined with chondroinductive biomaterials and bioactive factors (e.g., growth factors, peptides) for neotissue formation in cartilage focal defects. MSCs are increasingly employed for OA treatment. CARTISTEM^®^ (Medipost), approved by South Korea’s Ministry of Food & Drug Safety, provides allogeneic umbilical cord blood-derived MSCs with HA for OA treatment [28]. Recently, JointStem^®^ (RNL), consisting of adipose tissue-derived MSCs, entered Phase I/II clinical trial designed to treat patients with degenerative arthritis. Although the effectiveness of MSC therapy has been demonstrated in numerous pre-clinical and clinical studies, the mechanism of action is still unclear. MSC may exert therapeutic effect by engraftment or the release of trophic mediators. Unlike smooth normal cartilage, the OA cartilage surface is rough with exposed collagen fibrils that facilitate cell adhesion and engraftment; engrafted MSCs can form cartilage neotissue. Additionally, MSCs secrete a range of trophic factors, cytokines, and neuroregulatory peptides, which can stimulate endogenous tissue repair and inhibit inflammation. 

Embryonic stem cells (ESCs) and induced pluripotent stem cells (iPSCs) are pluripotent stem cells that are also employed in cartilage tissue engineering for their chondrogenic differentiation capacity [29,30]. However, ethical concerns, low yield of target cells, and other important limitations deter their use in OA stem cell therapy, even for research purposes. Capable of differentiation to virtually any differentiated cell type, ESCs and iPSCs experience heterogeneous and uncontrolled differentiation. This restricts direct transplantation of ESCs or iPSCs into a focal defect or OA joint for concerns of therapeutic failure and more seriously, tumorgenicity. Indeed, Hwang et al., demonstrated that ESCs expanded in co-culture with chondrocytes resulted in more homogeneous ESC-derived cartilage tissue both in vitro and in vivo [31]. Some studies have generated MSCs derived from ESCs and utilized their exosomes [32] and spheroids [33] in murine and primate OA models. Similarly, exosomes secreted by iPSC-derived MSCs were used in regenerating OA in a rat model [34]. Although ESC- or iPSC-derived MSCs add a level of complexity and sophistication to research, the advantages of using ESC- or iPSC-derived MSCs (as opposed to MSCs) cannot outcompete the disadvantages.

## 2. Biomaterials for OA Repair Applications 

### 2.1. Hydrogels for Cartilage Tissue Regeneration

Hydrogels are cross-linked 3D polymer networks with high water content. In tissue engineering, hydrogels are used alone or in conjunction with cells for biomedical applications. Hydrogels can be fabricated from natural, synthetic, or a mix of these polymers. In cartilage tissue engineering, cartilage ECM-derived or -mimetic biomaterials are used often to build a regenerative environment for chondrocytes and MSCs. HA, chondroitin sulfate (CS), and collagen are the main components of the cartilage extracellular matrix. Other natural polymers such as gelatin, alginate, and chitosan, are also readily used. However, most naturally derived polymers are mechanically weak and undergo rapid degradation. Therefore, biodegradable and biocompatible synthetic polymers such as poly(ethylene glycol) (PEG), polyvinyl alcohol (PVA), and poly(DL-lactic-co-glycolic acid) (PLGA) are also commonly used for cartilage tissue engineering. These synthetic polymers are generally degraded by hydrolysis of covalent bonds; resistance to hydrolytic cleavage varies, depending on steric hindrance at the cleavage site. For example, polycaprolactone can take as long as 24 months to undergo complete degradation [35]. Moreover, they have extensive possibilities for chemical modification to tune the hydrogels’ biological (e.g., conjugation of cell-adhesive peptides), physical (e.g., pore size and swelling ratio), and mechanical properties. Increasingly, researchers are endowing functionality to hydrogels such as injectability and adhesiveness to facilitate use in cartilage tissue regeneration. This section will discuss various injectable and adhesive hydrogels, their fabrication methods, and chondrogenic effects in detail.

#### 2.1.1. Injectable Hydrogels

Injectable hydrogels are particularly useful for cartilage tissue engineering given their adaptability to defect shape and size and the minimally invasive way of in vivo delivery. Injectability of hydrogels is enabled by in situ polymerizations via enzyme-mediated crosslinking, Schiff base crosslinking, photocrosslinking, and thermosensitive polymers.

Enzyme-mediated crosslinking utilizes enzyme-catalyzed chemical bond formation to crosslink polymers under mild physiological conditions. Substrates of the enzyme may be conjugated onto the polymer backbone prior to crosslinking, if not already present on the polymer. Horseradish peroxidase (HRP) and tyrosinase are commonly used in enzyme-mediated crosslinking by catalyzing the coupling of phenol derivatives (e.g., tyramine) (Figure 3). Jin et al., produced a dextran-hyaluronic acid hydrogel (HA-g-DX-TA) where dextran-tyramine was conjugated onto HA to mimic the molecular structure of proteoglycans present in the native cartilage ECM [36]. Enzymatic cross-linking of tyramine residues rapidly formed hydrogels within two minutes. HA-g-DX-TA hydrogels enhanced chondrocyte proliferation and ECM production at three weeks of in vitro culture compared to DX-TA hydrogels. Lee et al., synthesized gelatin-poly (ethylene glycol)–tyramine (GPT) graft copolymer, incorporating platelet-rich plasma (PRP) as a bioactive factor [37]. Derived from blood, PRP is a rich mix of anabolic growth factors including platelet-derived growth factor (PDGF), epidermal growth factor (EGF), insulin-like growth factor-1 (IGF-1), transforming growth factor-beta (TGF-β), and basic fibroblastic growth factor (bFGF). The chondrocyte-encapsulated hydrogel was studied in both the in vitro and in vivo rabbit osteochondral defect model. PRP-carrying GPT hydrogels showed an immediate increase in the expression of cannabinoid receptor, CB1, and CB2, which have been reported to have anti-inflammatory and analgesic effects in animal arthritis models. Injection into osteochondral defects indicated more cartilage regeneration in PRP-carrying GPT hydrogels. In another study, Teixeria et al., similarly supplemented dextran–tyramine hydrogels with platelet lysate and demonstrated enhanced proliferation, cell migration (chemotaxis), and chondrogenesis of encapsulated MSCs in vitro [38].

Schiff base crosslinking is achieved by the reaction between an alcohol, amine, or hydrazide group with an aldehyde. The reaction takes place in mild, physiological conditions, eliminating the problem of cytotoxicity of encapsulating cells. Yan et al., fabricated injectable hydrogels from hydrazide-modified poly(l-glutamic acid) and aldehyde-modified alginate [39]. The hydrogel precursor and chondrocyte mix were subcutaneously injected into mice using a double syringe. At 12 weeks in vivo, the Safranin O and Toluidine Blue stainings were deeper in color and more ECM accumulation, demonstrating better cartilage tissue formation compared to cell only injection. In another study, Cao et al., utilized the Schiff base reaction between the amino group of glycol chitosan and the aldehyde group of a PEG analog to encapsulate chondrocytes (Figure 4). Good viability and maintenance of the chondrocyte phenotype were observed after two weeks of in vitro culture [40].

Photocrosslinking utilizes a photoinitiator, which upon exposure to visible or UV light creates free radicals that initiate polymerization. Methacrylate or acrylate-conjugated polymers are commonly used for in situ photocrosslinkings. Kim et al., incorporated photoreactive methacrylates to a CS or HA backbone and mixed either ECM component with PEG [41]. Since PEG lacks cell-binding moiety and is biologically inert, a tripeptide composed of arginine, glycine, and aspartate (RGD) was conjugated to PEG to promote cell adhesion. Hydrogel precursors gelled within 5 min of UV exposure (3.5 mW/cm^2^) in the presence of photoinitiator Irgacure 2959. After three weeks of in vitro culture, bovine chondrocytes exhibited different behavior, depending on the presence of the ECM component (HA or CS). Compared to its HA-based counterpart, RGD-containing CS-based hydrogels indicated the most abundant glycosaminoglycan (GAG) and collagen production in histology and biochemical assays and the highest chondrogenic gene expression level in the real-time polymerase chain reaction. Bian et al., produced methacrylated HA hydrogels for MSC chondrogenesis [42]. Polymer concentration and UV (wavelength: 360 nm; intensity: 1.2 mW/cm^2^) exposure time were varied to evaluate the effect of crosslinking density on MSC chondrogenesis. Increased crosslinking by increasing either polymer concentration or light exposure time decreased the overall cartilage ECM content and restricted matrix diffusion. In addition, MSCs in more highly crosslinked HA hydrogels displayed hypertrophic differentiation and matrix calcification. While Irgacure 2959 is the most widely used synthetic photoinitiator, cytotoxicity limits its use to low concentrations of 0.05 wt.%. Another commonly used synthetic photoinitiator is lithium acylphosphinate (LAP),which initiates photopolymerization in visible light and is more desired than UV light for cell viability [43,44].

Recently, researchers have resorted to flavonoids as alternatives to synthetic photoinitiators. Consisting of two phenyl rings, flavonoids are capable of excitation via UV or visible light exposure. Riboflavin has demonstrated excellent potential as a safe and natural alternative. Methacrylate was conjugated to glycol chitosan and exposed to 5 min of visible blue light (wavelength: 400–500 nm; intensity: 300 mW/cm^2^) in the presence of riboflavin for gelation, which showed relatively high cell viability of encapsulated chondrocytes [45]. The addition of HA further promoted chondrocyte proliferation and cartilage ECM accumulation. Riboflavin can also aid in crosslinking collagen. Riboflavin-induced photocrosslinking of collagen was a technology first implemented in corneal crosslinking for keratoconus patients via UV-A radiation. The crosslinking reaction involves the production of singlet oxygen, which subsequently reacts with nearby groups. Tyrosine and histidine residues in collagen may be associated with this reaction [46]. In another study, Heo et al., fortified the natural physical crosslinking of collagen hydrogels with riboflavin-induced photocrosslinking (Figure 5). Although biologically active and biomimetic, collagen hydrogels are usually mechanically weak and undergo rapid degradation. Collagen hydrogel containing 0.01% *w*/*v* riboflavin showed a 5.5 times greater elastic modulus, less cell-mediated contraction, and slower enzymatic degradation [47]. Incorporation of crosslinked HA further enhanced the bioactivity of the hydrogel for chondrocyte ECM production as well as MSC chondrogenic differentiation in both the in vitro and in vivo subcutaneous implantation model [48].

Thermosensitive hydrogels use thermosensitive biomaterials alone or in conjugation with other biomaterials. Pluronic F127 and poly(*N*-isopropylacrylamide) (PNIPAM) are in a liquid state at low temperature and gels at near-physiological temperatures, enabling in situ polymerization. Jung et al., mixed heparin-conjugated Pluronic F127 and HA to synthesize a thermosensitive hydrogel that gelled within 10 min at 37 °C. Heparin was incorporated for the coupling of chondrogenic growth factor TGF-β1 to the hydrogel for sustained release. The hydrogel precursor solution was injected into the rabbit osteochondral defect. Alcian blue staining showed strong staining for cartilage ECM in the TGF-β1-conjugated group [49]. Another study added chitosan to thermosensitive components and HA. PNIPAM was first grafted to chitosan to form chitosan-*g*-PNIPAM (CPN), which was then grafted to HA to synthesize HA-CPN [50]. Compared to CPN hydrogels, both chondrocytes and meniscal chondrocytes demonstrated steadily increasing collagen and GAG accumulation over the course of in vitro culture (42 days), indicative of progressive cartilage tissue formation.

#### 2.1.2. Adhesive Hydrogel

In addition to injectability, adhesion to cartilage surface is a clear advantage by localizing treatment and decreasing the clearance of hydrogels. Adhesion can be enabled by electrostatic interactions, covalent bonds, and/or physical interactions. Balakrishnan et al., produced an oxidized alginate/gelatin hydrogel, which was crosslinked via a Schiff base reaction between aldehyde groups of oxidized alginate and amino groups of lysine residues in gelatin [51]. The adhesive property was attributed to the interaction between partially oxidized alginate and exposed collagen fibrils at the cartilage defect site. The adhesive strength was quantified by a custom-made burst test apparatus. In brief, a 4-mm incision was made in goat auricular cartilage tissue and sealed with the hydrogel; pressurized saline was injected into the sample and the pressure at which the gel burst was correlated with its tissue adhesiveness. The hydrogel’s burst pressure was 70 ± 3 mmHg, which was much lower than the intra-articular pressure in OA knees. Chondrocytes showed good cell viability and maintained their chondrogenic phenotype in the hydrogel as indicated by biochemical assays, Safranin-O staining, and type II collagen and aggrecan immunostaining. These results were enhanced by the incorporation of bioactive factors such as dexamethasone, CS, and platelet-derived growth factor.

In another study, Kim et al., fabricated an injectable and adhesive hydrogel based on gelatin and tyramine-conjugated HA [52]. Unlike previous studies, this study employed tyrosinase instead of HRP and hydrogen peroxide. Tyrosinase oxidizes tyramine’s catechol group to reactive quinones that can form covalent bonds with thiol, amine, and imidazole groups on gelatin and native tissues for hydrogel crosslinking and tissue adhesiveness. Adhesiveness to porcine cartilage tissue was quantified by a tack test; the hydrogel was sandwiched between two cartilage layers that were pulled in opposite directions at a rate of 2 mm/min. Meniscal chondrocytes, encapsulated in this hydrogel for three weeks, showed higher cartilaginous ECM accumulation and chondrogenic gene expression levels compared to cells in the gelatin-only hydrogel. Injectability and adhesiveness, along with chondrogenic property of the gelatin/HA-tyramine hydrogel, provide the robust potential for use in minimally invasive intra-articular delivery.

Jennifer Elisseeff’s lab fabricated a polymer–peptide system for OA treatment, consisting of a collagen-binding peptide and an HA-binding peptide conjugated on a linear PEG chain or 8-arm PEG [53,54]. Collagen-binding peptide adheres to the collagen on the cartilage surface for localization of the polymer–peptide system, while the HA-binding peptide binds to and localizes synovial HA at the cartilage surface. In vitro, adhesion of various HA-binding peptides was assessed and screened via quartz crystal microbalance with dissipation monitoring and isothermal titration calorimetry. Then, the selected HA-binding peptide was evaluated in vivo via intra-articular injection into the post-traumatic OA mouse model. 8-arm PEG conjugated with HA- and collagen-binding peptides showed the best tissue binding and joint retention with strong localization at cartilage damage sites where collagen fibers were exposed. Treatment effects of the polymer–peptide system on OA were evaluated by histology and behavioral tests. Safranin-O staining showed strong red staining for GAG, an indication of less cartilage deterioration in the treated group compared to the control. Relatively fast response time to thermal stimulus in the hot plate test and near-normal weight-bearing pattern were indicative of reduced pain and joint discomfort, and thus, recovery from OA.

### 2.2. Inflammatory-Modulating Biomaterials for OA 

#### 2.2.1. Symptom-Modulating Treatments 

Although once thought to be due to simple wear and tear, OA is dynamically driven by inflammatory mediators that play an important role in ECM degradation. It is expected that the combination of current anti-inflammatory therapies and biomaterials will effectively modulate inflammation and promote regeneration in OA cartilage. For example, NSAIDs can be delivered intra-articularly with injectable hydrogels to prevent fast clearance of the drugs. Studies delivered doxycycline and dexamethasone with high molecular weight or crosslinked HA hydrogels [55,56,57]. Audrey et al., encapsulated celecoxib, a COX-2 inhibitor, in a poly (ε-caprolactone-co-lactide)-b-poly (ethylene glycol)-b-poly (ε-caprolactone-co-lactide) (PCLA-PEG-PCLA) triblock copolymer hydrogel and observed drug release kinetics in normal horse knee. A triblock copolymer composed of hydrophobic and hydrophilic blocks is well-suited for loading and sustained and prolonged release of hydrophobic drugs, which is the case for most NSAIDs [58]. Synovial fluid samples were aspirated from treated horse knees for up to four weeks post-injection and processed for celecoxib quantification. A reasonably high local drug concentration was maintained for a prolonged time when delivered using this vehicle. Another method of hydrophobic drug delivery utilizes host–guest interactions between cyclodextrin (e.g., β-CD, γ-CD) and the hydrophobic drug [59]. Sustained drug delivery can also take the form of microspheres. Bedouet et al., produced degradable microspheres, in which ibuprofen was immobilized via poly(PLGA-PEG) dimethacrylate, a hydrolyzable covalent linkage. Initial burst release was suppressed and ibuprofen was slowly released with hydrolysis of the covalent bond [60]. Lipopolysaccharides-induced inflammation, a common in vitro inflammation model, was largely suppressed due to sustained release of ibuprofen; cartilage explants produced lower levels of PGE2.

#### 2.2.2. Disease-Modifying Treatments

Recent studies have aimed at disease modification using proteins or genetic materials that inhibit catabolic pathways and inflammatory cascade. Proteins, either inhibitors/antagonists or growth factors, are intra-articularly injected alone or loaded in nano- or microspheres for sustained release and slower clearance. Gene delivery to resident cells (chondrocytes and synovial MSCs), however, is a more fundamental approach. DNA is delivered via viral or non-viral vectors to increase transfection efficiency. RNAs such as miRNA, siRNA, etc. are utilized to silence undesired genes. This section will discuss various therapeutic proteins and genetic materials that modulate catabolic and anabolic processes involved in OA disease progression. IL-1, which initiates the inflammatory cascade by binding to IL-1R, is at the most upstream of the cascade and is the most common target of anti-inflammatory treatments. IL-1 receptor antagonist (IL-1Ra) is a natural antagonist that binds to and prevents IL-1R from activation by its ligand. Although endogenously produced, exogenous administration IL-1Ra proteins into the joint space have shown to effectively inhibit inflammation. Rather than direct injection, studies have employed microspheres [61] or self-assembling nanoparticles that encapsulate [62] or have the protein tethered on the surface [63,64]. Whitmire et al., and Wang et al., confirmed vehicle-mediated IL-1R retention in normal rat knee joints, while Elsaid et al., observed the disease-modifying effect of PLGA microsphere-encapsulated IL-1R in the rat anterior cruciate ligament transection (ACLT) OA model. Exogenous IL-1Ra can also be supplied by gene transfer, virally or non-virally. Given its apparent lack of pathogenicity and very mild immune response, adeno-associated virus (AAV) is used to efficiently deliver the IL-1Ra gene in vivo [65,66]. Non-viral methods of IL-Ra gene transfer utilize chitosan-based nanoparticles as vectors to transfect chondrocyte and synovial MSCs [67,68,69]. Therapeutic DNA can be complexed to positively charged macromolecules such as chitosan via electrostatic interaction [69].

Another important upstream pro-inflammatory cytokine is TNF-α. Commercially available biological drugs such as infliximab and adalimumab are monoclonal antibody-based TNF-α inhibitors that bind to TNF-α and decrease its enzymatic activity. These commercially available TNF-α inhibitors are currently injected alone, but to avoid multiple injections, sustained antibody delivery can be achieved by immobilizing antibodies on a high molecular weight polymer such as hyaluronic acid via amide bond formation [70]. Associated with TNFR and IL-1R activation, transcription factor NF-κB is a pivotal mediator of inflammatory responses that induce the expression of inflammatory genes. Micro RNA (miRNA) and small interfering RNA (siRNA) have been employed to suppress NF-κB activities. These classes of RNAs silence the expression of specific genes by complementary binding and degradation of their mRNA. Several studies have incorporated NF-κB siRNA in nanoparticles fabricated from cell-penetrating peptide [71] and injectable thermoresponsive hydrogel for on-demand release [72]. 

IL-4 and IL-10 are anti-inflammatory cytokines that can reverse the effects of OA. IL-4 inhibits classical macrophage activation into pro-inflammatory M1 cells and promotes their activation into anti-inflammatory, pro-regenerative M2 cells. IL-10 represses IL-1 and TNF-α expression of M1 cells. Exogenous IL-4 can be supplied by AAV-mediated gene transfer [73]. Fellowes et al., utilized cationic liposomes to deliver the IL-10 gene to mice OA model [74]. Cationic liposomes, like chitosan, interact electrostatically with DNA, thus well-suited for DNA delivery. Since IL-4 and IL-10 are proteins like TNFR and IL-1R, they can be incorporated into hydrogel systems for sustained release. For example, heparin-conjugated star-shaped PEG hydrogel (starPEG-heparin) can be used for IL-4, since IL-4 has been shown to strongly bind to heparin [75]. LPS-challenged primary murine macrophages exhibited the M2 phenotype when treated with IL-4 releasing starPEG-heparin in vitro. Soranno et al., fabricated an HA-based self-assembling hydrogel for IL-10 sustained delivery, which was capable of mitigating local and systemic proinflammatory cascade in an acute kidney injury mouse model [76]. Although these studies do not directly target OA, such findings hold great potential for application in OA.

#### 2.2.3. Next-Generation of Immune-Modulating Hydrogels

Current tissue engineering approaches to OA utilize anti-inflammatory and/or anabolic factors alone or with hydrogels. In this context, hydrogels, which may or may not promote chondrogenesis, are merely carriers that optimize the delivery of the aforementioned factors. Some existing biomaterials exhibit limited anti-inflammatory properties. For example, low molecular weight xanthan gum downregulates caspase 3 [77], and heparin/heparin sulfate [78,79] and sulfated alginate can capture free IL-1β, suppressing IL1β-mediated inflammation [80,81]. Carboxymethylated chitin was reported to reduce MMP-1 expression in the rabbit OA model [82]. However, there are no hydrogels for OA that target the immune system at its essence. Next-generation hydrogels that modulate immune cells, particularly macrophages that function as the first line of defense, can lay the groundwork for potent OA immunotherapy.

The following are some immunomodulatory biomaterials that are used for non-cartilaginous engineering applications, which can be extended to OA treatment. Wendy Liu’s lab examined the role of cell shape on macrophage polarization and applied this knowledge to scaffold design for wound healing. Quiescent macrophages become activated into proinflammatory M1 macrophages under the classical pathway, or alternatively, into anti-inflammatory M2 macrophages. Upon observation of cell morphology and cytoskeleton organization, M2 macrophages exhibited elongated shape. Based on this finding, the Liu lab fabricated micropatterned grooves to directly control macrophage shape and thus, its polarization to M2 [83,84]. Findings on groove dimensions that induce M2 macrophage polarization may be extended to micropatterned hydrogels that favorably modulate macrophage polarization for OA treatment. In another study, Chen et al., fabricated a 3D-printed scaffold from a composite bioink composed of cartilage ECM and MSC exosomes for osteochondral defect regeneration in rabbits [85]. Chen et al., found that cartilage ECM not only promoted cell migration into the scaffold, but also induced M2 macrophage polarization of infiltrated macrophages when subcutaneously implanted into mice. Finally, when implanted into the osteochondral defect, the synovial membrane exhibited M2 markers in the presence of cartilage ECM. The use of cartilage ECM to promote M2 polarization can be further applied to osteoarthritis treatment.

Another approach to immune modulation mimics pathways through which T cells differentiate into Th2 cells, which produce anti-inflammatory cytokines IL-4, IL-10, and IL-13. Naïve CD4+ T cells interact with antigen-presenting cells (APCs) by recognizing: (1) antigen that binds to T cell receptor and (2) co-stimulatory molecules. T cell receptor stimulatory and co-stimulatory molecules are conjugated to biomaterial cores to fabricate artificial APCs [86]. Various materials such as polystyrene beads [87], PLGA microparticles [88], iron/dextran nanoparticles [89], and liposomes [90] have been functionalized with appropriate signals to mimic APCs. Artificial APCs promote differentiation and proliferation of Th2 cells. Jennifer Elisseef’s lab utilized tissue ECM-derived scaffolds to drive Th2 differentiation, IL-4 expression upregulation, and IL-4-stimulated macrophage polarization into M2 [91].

## 3. Conclusions

With elderly and obese populations growing, the incidence and prevalence of OA are rapidly increasing and the socioeconomic burden related to this debilitating, but not fatal disease is expected to rise. Current clinical therapies concern symptom modification by NSAID injections and viscosupplementation. MSC intra-articular injection has shown positive results in many studies. The mechanism of action is still unclear, whether it is a result of MSC engraftment on damaged cartilage surface or the release of trophic factors that inhibit inflammation and stimulate endogenous tissue repair. This review covered functional hydrogels, which not only promote cartilage regeneration to reverse OA-induced damage, but can also act as delivery carriers of NSAIDs and proteins that target upstream effectors of the inflammatory cascade. Immune-modulating biomaterials that induce M2 macrophage polarization are highly anticipated next-generation biomaterials for disease-modifying OA treatments.

## Figures and Tables

**Figure 1 cells-09-00419-f001:**
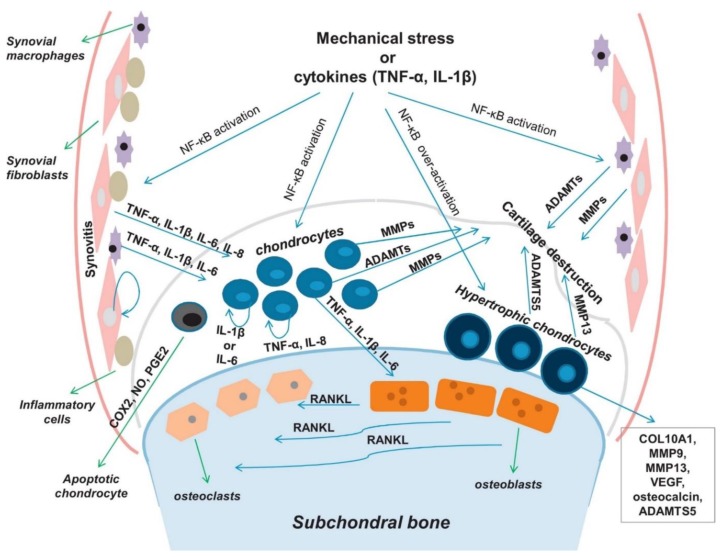
OA pathophysiology. NF-κB signaling network governs OA development and progression. Reproduced with permission from Rigoglou, S. et al., Biochem Cell Biol; published by Elsevier, 2013.

**Figure 2 cells-09-00419-f002:**
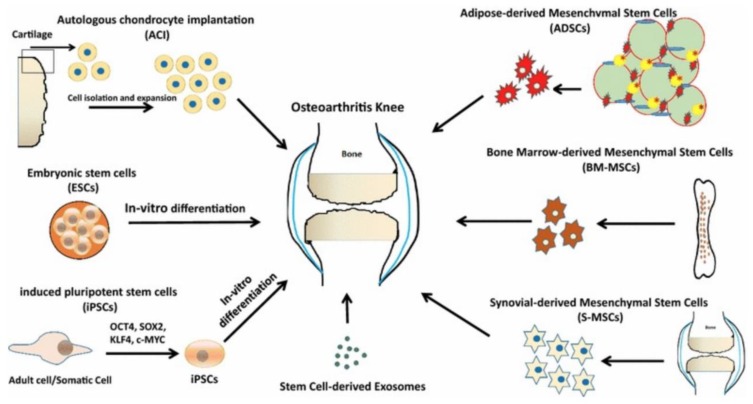
Cell-based treatments for OA. Schematic diagram illustrating the current clinical approaches to cell-based therapy for cartilage tissue engineering. Reproduced with permission from Burke, J. et al., Clin Transl Med; published by SpringerOpen, 2016.

**Figure 3 cells-09-00419-f003:**
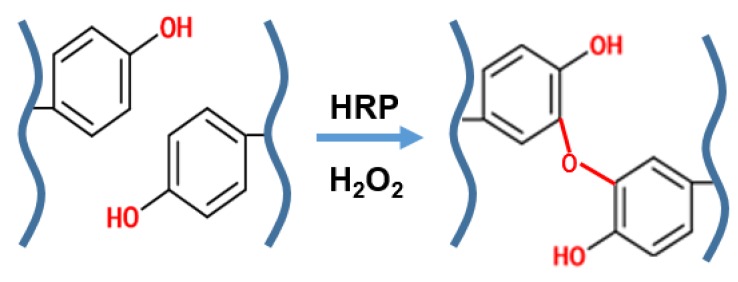
Enzyme-mediated crosslinking. Scheme of enzyme-mediated crosslinking of tyramine residues. Oxidation of tyramine residues enables stable bond formation.

**Figure 4 cells-09-00419-f004:**
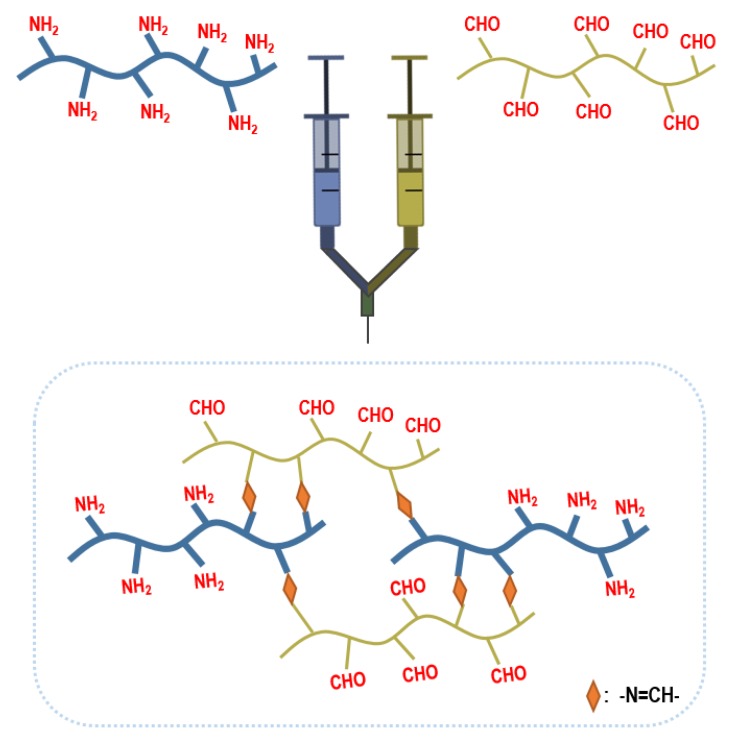
Schiff base crosslinking. Schiff base reaction between the amine group of glycol chitosan and the aldehyde group of PEG analog.

**Figure 5 cells-09-00419-f005:**
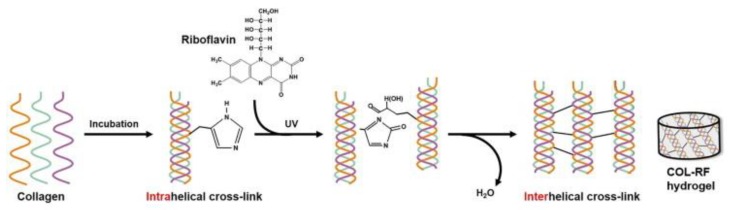
Riboflavin-induced photocrosslinking. Riboflavin acts as a photoinitiator of the reaction between potentially reactive R groups of amino acids in collagen chains such as histidine and tyrosine. Reproduced with permission from Koh, R.H. et al., Acta Biomaterialia; published by Elsevier, 2017.

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
