# Peer review of "Inflammation-Modulating Hydrogels for Osteoarthritis Cartilage Tissue Engineering"

_cells, 2020, doi:10.3390/cells9020419_

Round 1
Reviewer 1 Report
The authors describe in this review some aspects of osteoarthritis physiopathology and its current management highlitghting also the limitations for each kind of treatment.
The review is well structured and although more focused on the use of hydrogels in the management of this condition I suggest to the authors to extend the section relative to cell based treatment, given that this aspect has assuming a more growing interest in the scientific community. The ACI is surely a theraupetic approach widely used but i think that other cell based approaches can be revised and duscussed.
Author Response
We appreciate your comment. Following the discussion on ACI, we already discuss MSC intra-articular injection for OA therapy(Page 7, lines 132-145). Additional information on this therapy has been added in the paragraph. Moreover, a paragraph on other types of stem cells (embryonic and induced pluripotent stem cells) has been added (Pages 7-8, lines 146-158).
Reviewer 2 Report
The review article by Koh et al. entitled, “Inflammation-modulating hydrogels for osteoarthritis cartilage tissue engineering” surveys the current status of hydrogel-based therapies for osteoarthritis (OA). Overall, the manuscript is informative and contains essential information of the rapidly growing field. The knowledge summarized in this manuscript will be a great addition to the multidisciplinary efforts to treating OA. Therefore, the manuscript is recommended for publication in the journal Cells. However, the manuscript needs minor revision to improve its quality and clarity. Below are the detailed instructions for revision.
Be consistent with the abbreviations. For example, osteoarthritis in its full name appears on the first page, then “osteoarthritis (OA)” appears in the next page. Mesenchymal stem cells were introduced earlier with its abbreviation (MSCs), then its full name mesenchymal stem cells without abbreviation appears later. Write the full name before abbreviations, such as RT-PCR (page 10), GAG (page 10), PCLA-PEG-PCLA (page 14) and MS (page 15). There are many spelling, grammar and stylistic errors throughout the manuscript. For example, allogenic -> allogeneic (page 7), mimics -> mimic (page 8), encapsulated -> encapsulating (page 8), gelated -> gelled? (page 12), researches -> research (page 15), to name a few. The entire section 2.2.1 should be re-written as there are too many grammar errors. The manuscript begins with the definition of cartilage, but what is explained is articular cartilage (“acts as a cushion between the bones…”). Please check and correct it. Also, it will be better to provide more details about the cartilage (e.g. composition, resident cell types) at the beginning of the manuscript (in section 1.1). In its current form, such information is spread throughout the article (e.g. section 2.1). In the discussion of synthetic polymers for hydrogels (page 9), please comment on the degradation rates, as the authors mentioned the rapid degradation of natural polymers being a major limitation. When the authors introduced “Enzyme-mediated crosslinking”, they described as if HRP is the only enzyme that is used for hydrogel. (page 9) However, HRP is one of many enzymes that can be used to make hydrogels. The authors also mention tyrosinase later in the manuscript. In the discussion of photoinitiators, please be more specific about the wavelengths that are used. The authors introduced flavonoids as alternatives to Irgacure 2959. It will be beneficial to include other synthetic photoinitiators that utilize visible light, such as LAP. In the discussion of anti-inflammatory cytokines (IL-4 and IL-10), please make some connection with hydrogels since this review paper is about hydrogel-based treatments of OA. There may not be prior examples of incorporating these cytokines in the hydrogels, but it may be more useful for the readers if the authors provide the potential application of hydrogels in this aspect. The same goes with an example of M2 macrophage (page 17). How is it (or can it be) related to hydrogels? The authors should be able to make some brief comments on this. A concluding paragraph is missing.Author Response
Please see the attachment.

Reviewer 3 Report
Koh and co-authors, in their review article provide a good summary of current and future immunomodulatory therapies, focusing mainly consisting on the use of hydrogels having anti-inflammatory effects. This review is relevant in the field of osteoarthritis.
The reviewer enjoyed reading this paper but realized that the following modifications/corrections are required
A. Koh and co-authors must indicate a short Conclusion section that includes a summary and the authors’ comments on current approaches and future directions
B The authors must clearly indicate if the cited studies were performed in vitro or in vivo and the models used (these important info are often not provided). See some examples:
2.2.1. Symptomatic modulating treatments (Lines 298-299). “this system was well tolerated after intra-articular administration …53”: which animal model was used in this study?
2.2.1. Symptomatic modulating treatments (Lines 304-305). “This study represents the incorporation of ibuprofen prodrug within a PEG-hydrogel … 54” in vivo or in vivo? Which model?
2.2.2. Disease-modifying treatments (Lines 322-323). “Although endogenously produced, exogenous administration IL-1Ra proteins into the joint space…”: of patients or animals? (Which animal?)
C. Several sentences are not clearly written (or difficult to read) or uncorrected and must be rephrased. See some examples below:
Abstract (lines 17-18). “These therapeutics have been delivered alone or increasingly in delivery vehicles for sustained release and slower clearance of deliverables”: repetitions must be removed
Abstract (lines 22-23). Examples: “Besides inhibiting inflammation, anabolic and proliferative levels of OA chondrocytes are low ..”: it is not completely true considering that cell proliferation in OA cartilage is even enhanced in the early stages of the disease.
2.1.2. Adhesive hydrogel Introduction (lines 281-283). “Safranin O staining indicated strong red staining for GAG, indicative of ..” repetitions must be removed.
2.1.2. Adhesive hydrogel Introduction (line 260). “Another study by Kim et al.. fabricated an …” This sentence (like many similar other through the paper) must be modified as follow: “In another study, Kim et al fabricated an ..”
D. 1.3.3. Cell-based treatments (Line 121). The authors wrote that “The basic cell-based OA treatment is autologous chondrocyte implantation (ACI)”. It is however to highlight that in several countries OA is considered a contraindication for (matrix Assisted) ACI.
Round 2
Reviewer 1 Report
The authors implemented the manuscript as suggested.
Reviewer 2 Report
The revision looks good for publication.
Reviewer 3 Report
The authors have extensively and satisfactorily addressed all issues raised in my previous report.